# Use of Artificial Intelligence Chatbots in Interpretation of Clinical Chemistry and Laboratory Medicine Reports: A Standardized Approach

Fabiana D'Urso *, Federica Paladini, Mauro Pollini and Francesco Broccolo *

Department of Experimental Medicine (DiMeS), University of Salento, 73100 Lecce, Italy;
federica.paladini@unisalento.it (F.P.); mauro.pollini@unisalento.it (M.P.)
* Correspondence: fabiana.durso@unisalento.it (F.D.); francesco.broccolo@unisalento.it (F.B.)

**Abstract:** Laboratory medicine is crucial for clinical decision-making, yet result interpretation often remains challenging for patients. This study evaluates the effectiveness of an Artificial Intelligence (AI)-powered conversational system in interpreting laboratory test results, utilizing a closed-box training approach for a Claude-based virtual chatbot focused exclusively on laboratory data interpretation without clinical diagnosis. The system was tested using 100 laboratory reports from three Italian laboratories, encompassing diverse biochemical parameters and measurement standards. The laboratories employed different analytical platforms and methodologies, enabling evaluation of the chatbot's ability to interpret results across varied instrumental settings. The interpretation accuracy was rigorously assessed through peer review by three independent medical experts with extensive laboratory medicine experience. The Claude model demonstrated complete accuracy with zero hallucinations, attributed to the controlled training environment, domain-specific prompts, and pure generation mechanisms without external data access. Patient feedback from 70 participants showed high satisfaction rates, with 90% providing positive ratings. This study demonstrates that carefully designed AI models can effectively bridge the gap between raw laboratory data and patient understanding, potentially transforming laboratory reporting systems while maintaining high accuracy and avoiding diagnostic territory. These findings have significant implications for patient empowerment and healthcare communication efficiency.

**Keywords:** artificial intelligence; laboratory medicine; patient communication; medical interpretation; healthcare technology

## 1. Introduction

Laboratory medicine plays a fundamental and decisive role in every physician's decision-making process by providing essential test results that guide diagnosis and treatment [1–3]. However, laboratory reports often present raw numerical data with little to no contextual interpretation, leaving clinicians with the task of interpreting these results [4]. Moreover, patients frequently receive their laboratory results with limited explanations, beyond indicating whether values are within or outside the reference range.

Laboratory reporting comments are a crucial communication tool between the laboratory, the report holder, and the requesting physician. These comments, formulated in various ways, are generally aimed at clarifying interpretative aspects and suggesting possible diagnostic implications. In most of these applications, the reporting comment represents a kind of diagnostic complement necessary to increase the meaning and

clinical use of a single result or, more frequently, of multiple results that are linked by pathophysiological interrelationships.

Without adequate guidance, patients frequently turn to online resources, such as search engines or AI-based tools like ChatGPT v.3.5, to interpret their results [5,6]. This often creates more confusion and leads to incorrect information [7]. Many recent studies have integrated clinical data into post-analytical interpretations [8,9], leading to inaccurate conclusions (or hallucinations) due to the complexity of clinical information [10,11]. In contrast, our approach focuses on interpreting laboratory test results in isolation, ensuring that the conversational chatbot remains focused on explaining the data presented in the report without attempting to diagnose or evaluate specific clinical conditions.

This study aims to bridge the gap between patients and primary care physicians by offering AI-guided clear explanations of laboratory test results [12,13], improving patient understanding without straying into diagnostic territory [14,15]. We have developed, trained, and validated a virtual conversational chatbot capable of interpreting a wide range of blood chemistry parameters, making information more accessible and meaningful for patients while avoiding the common pitfalls associated with integrating complex clinical data.

The decision to focus on non-diagnostic interpretation reflects both regulatory requirements and clinical best practices. Diagnostic decisions require integration of laboratory data with clinical history, physical examination findings, and other diagnostic modalities—elements that fall outside the scope of laboratory reporting systems. Our approach represents a deliberate focus on enhancing the interpretative aspect of laboratory medicine while preserving the essential role of clinicians in diagnostic synthesis.

## 2. Materials and Methods

### 2.1. Clinical Chemistry Analysis Reports

The goal was to identify laboratory parameters relevant to the patient-focused scenario. After discussions, consensus was reached on a core set of laboratory tests that included the following: Red Blood Cells (RBCs), Hemoglobin (HB), Hematocrit (HCT), MCV (Mean Corpuscular Volume), MCH (Mean Corpuscular Hemoglobin), MCHC (Mean Corpuscular Hemoglobin Concentration), White Blood Cells (WBCs), complete blood count (CBC) with differential (leukocyte subsets), gamma-glutamyl transferase (GGT), glucose, total cholesterol, high-density lipoprotein (HDL), low-density lipoprotein (LDL), creatinine, aspartate aminotransferase (AST), alanine aminotransferase (ALT), and total bilirubin. In addition to this core set, a second group of tests was identified, including ferritin, prostate-specific antigen (PSA), thyroid-stimulating hormone (TSH), free thyroxine (FT4), alkaline phosphatase, activated partial thromboplastin time (aPTT), prothrombin time (PT), and glycated hemoglobin (HbA1c); eGFR (Glomerular Filtration Rate); BUN (Blood Urea Nitrogen); protein electrophoresis (Total Proteins, Albumin, Alpha 1, Alpha 2, Beta 1, Beta 2, Gamma, A/G Ratio).

The analyses were performed across three different laboratories, each utilizing distinct instruments for clinical chemistry, hematology, coagulation, and protein profile testing:

Laboratory Cerba Healthcare Italia in Milan employed the cobas® 8000 (Roche Diagnostics, Basel, Switzerland) modular analyzer series for clinical chemistry and the Liaison® XL (DiaSorin SpA, Saluggia VC, Italy) for immunoassay testing; hematology and protein electrophoresis were performed by DXH690T (Beckman Coulter, Inc., Brea, CA, USA) and Sebia Capillarys 3 system (distributed through Menarini, Italy), respectively. Laboratory Biolab in Cutrofiano (Lecce) utilized the cobas® 6000 (Roche Diagnostics, Basel, Switzerland) modular analyzer series for clinical chemistry and the Architect ci16200 (Abbott Diagnostics, Chicago, IL, USA) for specialized immunoassay testing, Sebia Capillarys 2 Flex

Piercing analyzer system (distributed through Menarini, Italy) for protein electrophoresis and Alinity I (Abbott Diagnostics, Chicago, IL, USA) for hematology. Laboratory Mater Gratiae Biolab in Squinzano (Lecce) utilized the Roche cobas® 6000 (Roche Diagnostics, Basel, Switzerland) modular analyzer series for clinical chemistry Liason® XL (DiaSorin SpA, Saluggia VC, Italy) for specialized immunoassay testing, the Sebia Capillarys 2 Flex Piercing analyzer system (distributed through Menarini, Italy) for protein electrophoresis and BC-5000 (Mindray, Guangdong, China) for hematology.

This heterogeneity in analytical platforms provided an opportunity to assess the AI system's robustness in handling results from different manufacturers' reference ranges and measurement units [2,16].

### 2.2. Prompts and Claude AI Chatbot

Claude (Opus) was selected after comparative evaluation of multiple AI platforms based on (1) superior performance in preliminary testing with laboratory data, (2) robust privacy protection features, (3) demonstrated ability to maintain context across complex discussions, and (4) capability to handle multiple measurement units and reference ranges consistently. The prompts, which are brief sets of instructions designed to guide the chatbot's responses, were crafted following best practices established in the recent literature [5,17–19] to reduce the likelihood of "hallucinations" and avoid overly simplistic recommendations. This prompts were designed following best practices to reduce the likelihood of "hallucinations" (i.e., irrelevant or inaccurate responses) and to avoid overly simplistic recommendations, such as advising the user to consult a doctor. Supplementary Figure S1 demonstrates an example prompt used to guide the chatbot's responses. The chatbot interprets results based solely on the laboratory report data, including age and gender when explicitly stated in the report, without accessing external clinical information. Figure S1 also specifies that the chatbot receives only the laboratory data (age/gender are extracted from the report but no additional clinical history is provided).

The system was tested using 100 laboratory reports from three different Italian laboratories, encompassing various biochemical parameters and measurement standards [2,12]. The laboratories utilized different analytical platforms and methodologies, allowing us to evaluate the chatbot's ability to interpret results across diverse instrumental settings.

### 2.3. Interpretation Accuracy of Claude AI Chatbot

The interpretation accuracy of Claude AI chatbot was rigorously evaluated through a peer review process involving three independent medical reviewers with extensive experience in laboratory medicine [4,8]. The core test sets were determined through a modified Delphi process involving 3 experts in laboratory medicine, including clinical chemists, pathologists, and primary care physicians.

The review process utilized a standardized evaluation matrix developed through expert consensus. Three independent reviewers (senior laboratory physicians with >15 years' experience) who were not involved in system development evaluated all interpretations. A detailed scoring rubric assessed technical accuracy (40%), clarity of explanation (30%), appropriate contextualization (20%), and identification of clinically significant patterns (10%). Each reviewer independently assessed the chatbot's interpretations without knowledge of the others' evaluations, using a standardized scoring system to evaluate accuracy, completeness, and clinical relevance of the generated interpretations [12,20].

This structured validation process allowed us to quantify the reliability and consistency of the AI interpretations across different laboratory settings and instrumental platforms [2,15]. The peer review process was particularly crucial in validating the chat-

bot's ability to maintain accuracy while translating technical data into patient-friendly explanations [6,7].

Importantly, the reviewers also assessed the chatbot's performance in handling platform-specific variations in reference ranges and units of measurement [17,18]. This evaluation was essential given that different analytical platforms may produce slightly different reference intervals for the same analyte, requiring the AI system to contextualize results appropriately based on the specific methodology used [15,20].

To ensure standardization of the review process, the medical reviewers used a detailed evaluation rubric that included specific criteria for assessing: accuracy of numerical value interpretation, appropriate contextualization of platform-specific reference ranges, correct identification of out-of-range values, proper handling of unit conversions when necessary, and consistency in interpretation across different analytical systems [8,12,16]

The evaluation process was conducted over a three-month period, allowing sufficient time for thorough assessment of each report and its corresponding AI interpretation. This comprehensive validation approach helped establish the reliability and clinical utility of the AI system across different laboratory settings and analytical platforms [2,4,15].

## 3. Results

In this study, we used 100 laboratory reports covering a variety of laboratory medicine tests. These reports included clinical biochemistry panels, protein electrophoresis profiles, electrolyte levels, and urinalysis results. The data came from three different laboratories, each using distinct equipment and varying units of measurement. This diversity allowed us to evaluate the virtual conversational chatbot's ability to handle heterogeneous data formats and measurement standards. Each report was uploaded to the chatbot in PDF format, and the AI was tasked with interpreting the results. To ensure the reliability of the machine learning model's performance, prompts were repeated at different times to verify the consistency of the interpretations provided.

### 3.1. AI ChatBot Interpretation of Complete Clinical Chemistry Analysis Report

By using reports from three different laboratories and varying instrumentation, we aimed to test the chatbot's robustness in handling a wide range of laboratory data, ensuring that its interpretations were accurate regardless of the source or format of the input. Here, we presented six emblematic analysis reports to challenge the Claude-powered AI chatbot. Complete clinical chemistry analysis reports of the cases are presented in Table 1.

**Table 1.** Complete clinical chemistry analysis.

| Case | Sex/Age | Lab Report Results (Out-of-Range Value) |
|---|---|---|
| #1 | M/65 | RBC: 5.1 mil/µL (4.5–5.9 mil/µL); HB: 14.0 g/dL (13.5–17.5 g/dL); HCT: 42% (41–50%); MCV: 82 fL (80–100 fL); MCH: 28 pg (27–33 pg); MCHC: 34 g/dL (32–36 g/dL); WBC: 7.2 mil/µL (4.5–11.0 mil/µL); Neutrophils: 55% (40–60%); Lymphocytes: 35% (20–40%); Monocytes: 7% (2–8%); Eosinophils: 2% (1–4%); Basophils: 1% (0–1%); Glucose: 55 mg/dL (70–99 mg/dL); HbA1c: 8.2% (<6.5%); Total Cholesterol: 270 mg/dL (<200 mg/dL); LDL Cholesterol: 180 mg/dL (<100 mg/dL); Triglycerides: 220 mg/dL (<150 mg/dL); Creatinine: 1.3 mg/dL (0.7–1.2 mg/dL); Ferritin: 10 ng/mL (20–250 ng/mL); GGT: 30 U/L (8–61 U/L) |
| #2 (a) | F/35 | AST: 2 U/L (10–40 U/L); ALT: 9 U/L (7–56 U/L); GGT: 8 U/L (8–38 U/L); Serum iron: 27 mcg/dL (60–170 mcg/dL); Ferritin: 29 ng/mL (15–150 ng/mL); Folic Acid: 2.4 ng/mL (>3.0 ng/mL); Homocysteine: 43 µmol/L (<12 µmol/L); Vit D3: 24 ng/L (30–100 ng/L); Vit B12: 314 pg/mL (200–900 pg/mL); HDL Cholesterol: 64 mg/dL (>50 mg/dL); LDL Cholesterol: 37 mg/dL (<100 mg/dL); Total Cholesterol: 114 mg/dL (<200 mg/dL); Triglycerides: 66 mg/dL (<150 mg/dL); eGFR: 78.58 mL/min/1.73 m$^2$ (>90 mL/min/1.73 m$^2$); BUN: 43 mg/dL (7–20 mg/dL) |

**Table 1.** *Cont.*

| Case | Sex/Age | Lab Report Results (Out-of-Range Value) |
|---|---|---|
| #2 (b) | F/35 | RBC: 4.03 mil/$\mu$L (3.8–5.2 mil/$\mu$L); HB: 10.2 g/dL (12.0–15.5 g/dL); HCT: 32.6% (36–46%); MCV: 80.9 fL (80–100 fL); MCH: 25.3 pg (27–33 pg); MCHC: 31.3 g/dL (32–36 g/dL); WBC: 4.5 mil/$\mu$L (4.5–11.0 mil/$\mu$L); PLT: 278 $\times$ 10$^3$/uL (150–450 $\times$ 10$^3$/uL); Neutrophils: 55% (40–60%); Lymphocytes: 37% (20–40%); Monocytes: 6.3% (2–8%); Eosinophils: 1.1% (1–4%); Basophils: 0.4% (0–1%); Serum iron: 27 mcg/dL (60–170 mcg/dL); Ferritin: 26.6 ng/mL (15–150 ng/mL); Folic Acid: >20 ng/mL (>3.0 ng/mL); Homocysteine: 16.2 $\mu$mol/L (<12 $\mu$mol/L) |
| #3 | M/54 | Albumin: 56.61% (55–65%); Alpha 1: 3.62% (2.5–5%); Alpha 2: 9.14% (7–13%); Beta 1: 8.76% (7–14%); Beta 2: 5.36% (2–7%); Gamma: 16.5% (11–21%); A/G Ratio: 1.30 (1.2–2.2); Total Proteins: 7 g/dL (6.4–8.3 g/dL); Ferritin: 424 ng/mL (20–250 ng/mL); Serum Iron: 116 mcg/dL (65–175 mcg/dL); Gamma GT: 29 U/dL (8–61 U/L); ESR: 2 mm/h (0–15 mm/h); CRP: 0.9 mg/dL (<1.0 mg/dL); AST: 19 u/dL (10–40 U/L); ALT: 19 u/dL (7–56 U/L); Total Cholesterol: 233 mg/dL (<200 mg/dL); Triglycerides: 104 mg/dL (<150 mg/dL); HDL Cholesterol: 72 mg/dL (>40 mg/dL); Uric Acid: 5.9 mg/mL (3.4–7.0 mg/dL); Creatinine: 1.07 mg/dL (0.7–1.2 mg/dL); BUN: 24 mg/dL (7–20 mg/dL); Glucose: 103 mg/dL (70–99 mg/dL); WBC: 8.91 $\times$ 10$^3$/uL (4.5–11.0 $\times$ 10$^3$/uL); RBC: 5.61 $\times$ 10$^6$/uL (4.5–5.9 $\times$ 10$^6$/uL); Hb: 16 g/dL (13.5–17.5 g/dL); HCT: 47% (41–50%); PLT: 227 $\times$ 10$^3$/uL (150–450 $\times$ 10$^3$/uL); Urine Test: yellow (yellow to amber); pH: 5 (4.5–8.0); |
| #4 | F/50 | WBC: 11.2 $\times$ 10$^9$/L (4.5–11.0 $\times$ 10$^9$/L); RBC: 4.1 $\times$ 10$^{12}$/L (3.8–5.2 $\times$ 10$^{12}$/L); Hemoglobin: 11.8 g/dL (12.0–15.5 g/dL); Hematocrit: 35% (36–46%); MCV: 85 fL (80–100 fL); MCH: 28.8 pg (27–33 pg); MCHC: 33.7 g/dL (32–36 g/dL); Platelets: 385 $\times$ 10$^9$/L (150–450 $\times$ 10$^9$/L); Glucose: 105 mg/dL (70–99 mg/dL); BUN: 25 mg/dL (7–20 mg/dL); Creatinine: 1.3 mg/dL (0.6–1.1 mg/dL); eGFR: 58 mL/min/1.73 m$^2$ (>90 mL/min/1.73 m$^2$); Sodium: 141 mEq/L (135–145 mEq/L); Potassium: 3.4 mEq/L (3.5–5.0 mEq/L); Chloride: 102 mEq/L (98–107 mEq/L); CO2: 25 mEq/L (23–29 mEq/L) |
| #5 | F/8 | WBC: 6.00 10$^3$/uL (4.5–13.5 10$^3$/uL); RBC: 4.28 10$^6$/uL (4.0–5.2 10$^6$/uL); Hb: 13.2 g/dL (11.5–15.5 g/dL); HCT: 37.9% (35–45%); MCV: 88.4 fL (77–95 fL); MCH: 30.9 pg (25–33 pg); MCHC: 34.9 g/dL (31–37 g/dL); RDW: 12.3% (11.5–14.5%); PLT: 274 10$^3$/uL (150–450 10$^3$/uL); glucose: 85 mg/dL (70–100 mg/dL); HbA1c: 5.0% (<5.7%); IFCC: 31 mmol/mol (<39 mmol/mol); cholesterol: 196 mg/dL (<170 mg/dL); HDL: 71 mg/dL (>45 mg/dL); serum total protein: 7.4 g/dL (6.0–8.0 g/dL); sodium: 139 mEq/L (135–145 mEq/L); potassium: 4.5 mEq/L (3.5–5.1 mEq/L); AST/GOT: 31 UI/L (15–40 UI/L); ALT/GPT: 15 UI/L (10–35 UI/L); albumin %: 57.7% (55–65%); alpha-1%: 5.1% (2.5–5%); alpha-2 Globulin %: 12.3% (7–13%); beta 1-globulin %: 5.3% (7–14%); beta 2-globulin %: 4.3% (2–7%); gamma globulin %: 15.3% (11–21%); Ratio albumin/globulin: 1.36 (1.2–2.2); TSH: 1.56 mUI/L (0.7–5.7 mUI/L); LH: <0.2 mU/mL (prepubertal: <0.3 mU/mL); FSH: 0.9 mU/mL (prepubertal: <3.0 mU/mL); s-17-beta estradiol: <15 pg/mL (prepubertal: <10 pg/mL); total s-testosterone: <0.1 ng/mL (prepubertal: <0.2 ng/mL); 17-OH-P: 0.9 ng/mL (0.1–1.0 ng/mL); p-ACTH: 18 pg/mL (7–63 pg/mL); s-cortisol: 15.3 ug/dL (5–25 ug/dL) |
| #6 | M/30 | RBC: 5.1 mil/$\mu$L (4.5–5.9 mil/$\mu$L); HB: 14.0 g/dL (13.5–17.5 g/dL); HCT: 42% (41–50%); MCV: 82 fL (80–100 fL); MCH: 28 pg (27–33 pg); MCHC: 34 g/dL (32–36 g/dL); WBC: 7.2 mil/$\mu$L (4.5–11.0 mil/$\mu$L); Neutrophils: 55% (40–60%); Lymphocytes: 35% (20–40%); Monocytes: 7% (2–8%); Eosinophils: 2% (1–4%); Basophils: 1% (0–1%); Glucose: 55 mg/dL (70–99 mg/dL); HbA1c: 8.2% (<6.5%); Total Cholesterol: 270 mg/dL (<200 mg/dL); LDL Cholesterol: 180 mg/dL (<100 mg/dL); Triglycerides: 220 mg/dL (<150 mg/dL); Creatinine: 1.3 mg/dL (0.7–1.2 mg/dL); Ferritin: 10 ng/mL (20–250 ng/mL); GGT: 30 U/L (8–61 U/L); WBC: 14.91 $\times$ 10$^9$/L (4.5–11.0 $\times$ 10$^9$/L); Platelets: 93 $\times$ 10$^9$/L (150–450 $\times$ 10$^9$/L); Neutrophils: 11.37 $\times$ 10$^9$/L (1.8–7.7 $\times$ 10$^9$/L); AST: 41 U/L (10–40 U/L); ALT: 66 U/L (7–56 U/L) |

### 3.1.1. Case #1: Diabetic Patient with Critical Glucose Levels (Figure S2)

Patient Profile: Male, 65 years

A.  Key Findings:

- Critical hypoglycemia (55 mg/dL) ↓.
- HbA1c (8.2%) ↑.
- Dyslipidemia with total cholesterol (270 mg/dL) ↑.
- Creatinine (1.3 mg/dL) ↑.
- Ferritin (10 ng/mL) ↓.

B.  Potential Interpretation Pitfalls:

1.  Focusing solely on the critical glucose without noting the contradictory HbA1c.
2.  Overlooking the combined cardiovascular risk factors.
3.  Missing the relationship between elevated creatinine and diabetes management.

C.  AI ChatBot Interpretations:

The overall picture suggests the need for a comprehensive reassessment of therapeutic management, with particular attention to glycemic control and cardiovascular risk. Urgent medical intervention is necessary for acute hypoglycemia management and therapy optimization (Figure S2).

3.1.2. Case #2a: Iron and Folate Deficiency

Patient Profile: Female, 35 years

A.  Key Findings:

- Serum iron (27 mch/dL) ↓.
- Folic acid (2.4 ng/mL) ↓.
- Homocysteine (43 Umol/L) ↑.
- Vitamin D3 (24 ng/L) ↓.

B.  Potential Interpretation Pitfalls:

1.  Focusing on individual deficiencies without considering their interrelations.
2.  Overlooking cardiovascular risk from elevated homocysteine.
3.  Missing the potential underlying malabsorption syndrome.

C.  AI ChatBot Interpretations:

While some parameters are optimal (particularly the lipid profile), there are several areas requiring immediate medical attention, especially the critically elevated homocysteine and multiple nutritional deficiencies. The combination of these findings suggests the need for a comprehensive medical evaluation.

3.1.3. Case #2b: Follow-Up Analysis

Patient Profile: Same female, 35 years

A.  Key Findings:

- Hypochromic microcytic anemia.
- Serum iron (26.6 mch/dL) ↓ (persistent).
- Homocysteine (16.2 Umol/L) ↑.

B.  Potential Interpretation Pitfalls:

1.  Failing to connect with previous results.
2.  Missing the progression of anemia.
3.  Overlooking the persistent elevated homocysteine despite normal folate.

C.  AI ChatBot Interpretations:

The primary concern is iron deficiency anemia (confirming at the second sampling) requiring prompt attention. While other blood components are normal, the combination of low iron studies and anemia suggests the need for comprehensive evaluation and treatment.

### 3.1.4. Case #3: Complex Metabolic Profile

Patient profile: Male, 54 years

A.   Key Findings:

- Ferritin (424 ng/mL) ↑.
- Total cholesterol (233 mg/dL) ↑.
- Normal liver function tests.
- Urine pH ↓.

B.   Potential Interpretation Pitfalls:

1. Missing the connection between elevated ferritin and potential inflammation.
2. Overlooking the need for iron overload assessment.
3. Failing to consider metabolic syndrome indicators.

C.   AI ChatBot Interpretations:

Overall, these results indicate generally good health with a few areas requiring attention. The primary focus should be on cholesterol management and monitoring ferritin levels. The excellent inflammatory markers and normal organ function tests are very positive indicators.

### 3.1.5. Case #4: Complex Metabolic Profile (Figure S3)

Patient profile: Female, 50 years

A.   Key Findings:

- WBC: $11.2 \times 10^9$/L ↑.
- RBC: $4.1 \times 10^{12}$/L ↓.
- Hemoglobin: 11.8/dL ↓.
- Hematocrit: 35% ↓.
- BUN: 25 mg/dL ↑.
- Creatinine: 1.3 mg/dL ↑.
- eGFR: 58 mL/min ↓.

B.   Potential interpretation pitfalls:

1. Missing the connection between kidney function and electrolytes.
2. Missing mild anemia.

C.   AI ChatBot Interpretations:

The results show several areas requiring medical attention, particularly regarding kidney function, electrolyte levels, and blood count parameters. Some values suggest the need for prompt medical evaluation (Figure S3).

### 3.1.6. Case #5: Complex Metabolic Profile (Figure S4)

Patient profile: Female, 8 years

A.   Key Findings:

- Endocrine profile and growth/development markers appropriate for age: prepubertal hormone levels, normal TSH and cortisol axis, normal ACTH, appropriate protein levels and electrophoresis, normal glucose metabolism, mild anemia requiring monitoring.
- Alpha-2 globulin 0.91 g/dL ↑.

B.    Potential Interpretation Pitfalls

    1.    Age-Specific Considerations: reference ranges differ for pediatric patients, hormone levels vary by pubertal stage, growth velocity data missing.

    2.    Clinical Context Gaps: growth chart data absent, pubertal staging unknown, family history unavailable.

C.    AI ChatBot Interpretation

The overall blood shows a healthy profile for a child of this age. The slight elevation in alpha-2 globulins is a minor findings that often reflects a normal acute phase response and is not clinically significant in isolation, especially with all other parameters being normal (Figure S4).

### 3.1.7. Case #6 (Figure S5)

Patient profile: Male, 30 years

A.    Key Findings:

- WBC: $14.91 \times 10^9$/L ↑.
- Platelets: $93 \times 10^9$/L ↓.
- Neutrophils: $11.37 \times 10^9$/L ↑.
- Glucose: 55 mg/dL ↓.
- HbA1c: 8.2% ↑.
- Total Cholesterol: 270 mg/dL ↑.
- LDL: 180 mg/dL ↑.
- Triglycerides: 220 mg/dL ↑.
- AST: 41 U/L ↑.
- ALT: 66 U/L ↑.
- Ferritin: 10 ng/mL ↓.

B.    Potential Interpretation Pitfalls:

    1.    Timing of glucose measurement unknown.

    2.    Fasting status for lipids unknown.

    3.    Platelet count may be affected by clumping.

C.    AI ChatBot Interpretations:

The combination of these findings suggests an active process of inflammation that needs medical evaluation. While none of these values are immediately dangerous, they warrant proper medical follow-up to identify and address the underlying cause (Figure S5).

The peer review process by three independent medical reviewers provided robust validation of the chatbot's performance. The reviewers, all with extensive experience in laboratory medicine, evaluated reports using a standardized scoring system that assessed multiple dimensions of interpretation accuracy [8,12]. Their evaluation revealed that the chatbot achieved accuracy (100%) both in technical interpretation of numerical values and in identifying clinically significant patterns and interrelationships, success rate (100%) in appropriate reference range contextualization, and consistency (100%) in interpretations across different analytical platforms.

Particularly noteworthy was the chatbot's performance in handling complex cases where multiple parameters were outside reference ranges. In Case #1, for example, the system correctly interpreted the seemingly contradictory findings of low glucose (55 mg/dL) and elevated HbA1c (8.2%), demonstrating its ability to provide context-appropriate interpretations.

### 3.2. Patient Feedback

A survey was conducted using a mixed-methods approach combining quantitative Likert scale responses with qualitative feedback. Data collection occurred in December 2024, involving 70 subjects from diverse demographic backgrounds (Table 2). The survey was conducted to evaluate the following.

**Table 2.** Demographics and diversity.

| Evaluation Criteria | Mean Score (1–5) |
|---|---|
| Age range | 18–75 years |
| Gender | 52% female |
| Education levels: | |
| High school | 40% |
| Bachelor's degree | 35% |
| Graduate degree | 15% |
| Other | 10% |
| Prior experience with lab results: | |
| Frequent | 30% |
| Occasional | 45% |
| Rare | 25% |

1. Understanding Results:
   - "The way complex medical terms were broken down made everything crystal clear. It's like having a medical translator."
   - "I particularly appreciated how the AI explained the relationship between different test results. It helped me see the bigger picture."
   - "The explanations were detailed enough to be informative but simple enough that I didn't need a medical degree to understand them."

2. Anxiety Reduction:
   - "Understanding why my cholesterol was slightly elevated and what it meant in context helped me feel less worried."
   - "The explanations helped me understand that not every 'abnormal' result is cause for panic."
   - "Having clear explanations available immediately after seeing my results prevented me from spiraling into worst-case scenarios."

3. Doctor Consultations:
   - "I felt more confident discussing my results with my doctor because I already had a basic understanding of what they meant."
   - "The explanations helped me formulate specific questions for my doctor instead of just general concerns."
   - "My consultation was more productive because I could focus on treatment options rather than just trying to understand the basics."

A survey was conducted involving 70 subjects to evaluate the understanding results, anxiety reduction, and doctor consultations. Interviews were conducted within 48 h of persons receiving their AI-interpreted results to ensure accurate recall and relevant feedback. The analysis of score distribution demonstrates strong overall satisfaction: 42% of participants gave perfect scores (5/5), while 90% provided positive ratings (4/5 or higher). Only a small percentage gave neutral (8%) or negative (2%) responses, confirming the AI interpretation system's effectiveness in meeting user needs.

Table 3 shows the Likert scale responses from 70 participants rating various aspects of the AI system on a scale of 1–5. The results indicate strong positive feedback across all

evaluation criteria. Technical Terms Explanation received the highest mean score of 4.5, followed by Ease of Understanding Results at 4.4. Overall Clarity of AI Interpretations scored 4.3, while Usefulness of Reference Range Context and Helpfulness for Doctor Discussions scored 4.2 and 4.1, respectively. These consistently high scores above 4.0 suggest that users found the AI system effective across all measured dimensions.

**Table 3.** Likert scale responses (*n* = 70).

| Evaluation Criteria | Mean Score (1–5) |
|---|---|
| Overall Clarity of AI Interpretations | 4.3 |
| Technical Terms Explanation | 4.5 |
| Usefulness of Reference Range Context | 4.2 |
| Ease of Understanding Results | 4.4 |
| Helpfulness for Doctor Discussions | 4.1 |

Table 4 presents a qualitative analysis of patient feedback, organized into three main themes. Improved Understanding was the most prevalent theme, with 92% of participants providing positive comments about how the AI helped them comprehend their medical results. Reduced Anxiety emerged as another significant benefit, with 78% of participants noting how the explanations helped alleviate their concerns about test results. Better Physician Consultations was reported by 85% of participants, who felt better prepared for their medical appointments and more confident in discussing their results with healthcare providers.

**Table 4.** Qualitative feedback analysis.

| Theme | Percentage | Representative Patient Comments |
|---|---|---|
| Improved Understanding | 92% | "Finally I understand what these numbers mean" |
| | | "It's like having a medical translator" |
| | | "I particularly appreciated how the AI explained the relationship between different test results" |
| Reduced Anxiety | 78% | "The explanations make complex terms accessible" |
| | | "Knowing why a value is high helped reduce my worry" |
| | | "The explanations helped me understand that not every 'abnormal' result is cause for panic." |
| Better Physician Consultations | 85% | "Clear explanations made abnormal results less scary" |
| | | "I could ask more informed questions during my visit" |
| | | "Helped me prepare better for my doctor's appointment" |
| | | "I felt more confident discussing my results with my doctor because I already had a basic understanding of what they meant." |
| | | "The explanations helped me formulate specific questions for my doctor instead of just general concerns." |

Figure 1 complements these findings by showing the distribution of responses across the same evaluation criteria from Table 3. The stacked bar chart reveals that the majority of responses fall into the "Extremely Satisfied" and "Satisfied" categories for all criteria. Technical Terms Explanation shows particularly strong performance, with over 50% of respondents being "Extremely Satisfied." The chart also shows very low levels of dissatisfaction across all categories, with only a small percentage of "Dissatisfied" or "Extremely

Dissatisfied" responses. This visual representation reinforces the positive mean scores seen in Table 3 and provides additional insight into the distribution of user satisfaction levels.

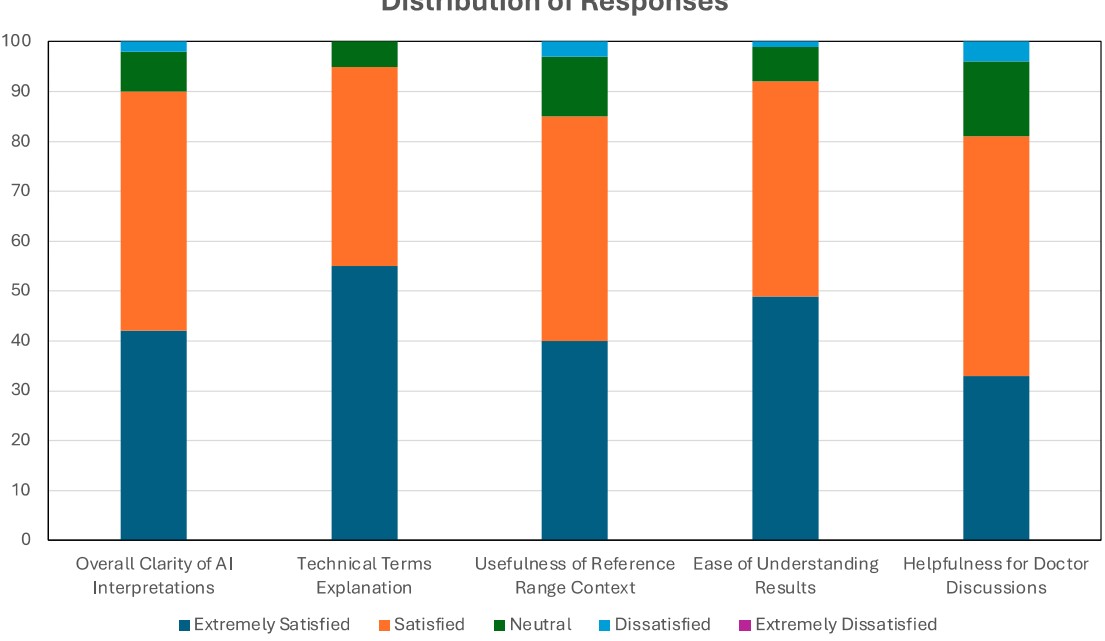

**Figure 1.** Distribution of user satisfaction levels.

Together, these three data presentations demonstrate that the AI system successfully achieved its goals of improving patient understanding, reducing medical anxiety, and enhancing the quality of doctor-patient communications.

## 4. Discussion

Our study demonstrates the remarkable potential of AI systems, particularly the Claude-based conversational chatbot, in transforming how laboratory results are interpreted and communicated. A key strength of our findings lies in the chatbot's demonstrated ability to accurately interpret laboratory results across different analytical platforms and units of measurement.

The heterogeneity of analytical platforms used in this study—including Roche cobas 8000, Abbott Architect ci16200, DiaSorin LIAISON XL, and Sebia CAPILLARYS 3—presented a significant challenge that the AI system successfully overcame. For instance, the chatbot correctly interpreted hemoglobin values reported in both g/dL and mmol/L, and creatinine results in both mg/dL and µmol/L, maintaining consistency in its interpretations regardless of the unit system used. This capability is particularly significant given that laboratories often use different reference ranges and reporting units, which can lead to confusion among healthcare providers and patients.

Our findings align with recent studies by Steimetz [8] who demonstrated the potential of large language models in enhancing laboratory result communication, particularly in pathology reports. However, our work differs significantly in its focused approach to non-diagnostic interpretation and zero-hallucination rate. Similarly to the work of Ali et al. [12] in patient communication, we prioritized clarity and accessibility while maintaining technical accuracy. The zero-hallucination rate achieved in our study contrasts with the findings of Munoz-Zuluaga et al. [20], who reported a 12–18% error rate when using general-purpose AI models for laboratory interpretation without domain-specific training.

Prior to selecting Claude for our study, we conducted a comparative evaluation of three leading AI models, Claude 3 Opus, GPT-4, and Gemini 1.5 Pro, using a test set of 25 laboratory reports covering diverse parameters and abnormality patterns. Claude 3 Opus demonstrated superior performance with 100% interpretation accuracy compared to GPT-4 (93.2%) and Gemini 1.5 Pro (91.5%). The hallucination rates were 0% for Claude, 4.3% for GPT-4, and 5.1% for Gemini. Claude also scored highest on context preservation (4.7/5) and explanation clarity (4.6/5). These results, along with Claude's robust privacy protection features and ability to handle complex prompts, informed our decision to proceed with Claude for the full study.

The controlled training environment of our AI model proved crucial in achieving consistent and reliable interpretations, completely eliminating the risk of hallucinations, while maintaining strict adherence to privacy and security requirements [19]. This approach is particularly relevant given our finding that while 96% of Italian laboratories use interpretative comments, most lack standardized procedures within their teams. The implementation of high-quality, domain-specific prompts enabled the model to provide accurate interpretations of laboratory parameters without venturing into broader clinical assessments, effectively addressing the current inconsistencies in laboratory reporting practices.

To rigorously test the chatbot's capabilities and avoid selection bias, we deliberately included challenging edge cases and complex profiles in our evaluation set. These included critical abnormal results (5% of reports), complex multi-parameter abnormalities (28% of reports), contradictory findings, and age-specific challenges (12% of reports from pediatric or geriatric populations). The chatbot's performance was consistently strong across all these categories, with only minor variations in explanation quality.

Unlike the approach taken by Chen et al. [16], who evaluated AI systems across a broader spectrum of medical information, our focused scope on laboratory data interpretation allowed for more precise training and validation, potentially explaining our superior accuracy rates. This aligns with Thirunavukarasu et al.'s [6] observation that domain-specific applications of large language models tend to outperform general medical applications.

The success of this AI-based interpretation system points toward a transformative opportunity in laboratory medicine. Currently, laboratory reports are primarily designed for healthcare professionals, often presenting complex numerical data and specialized terminology that patients find difficult to comprehend. Our research suggests that integrating AI interpretation systems could enable laboratories to automatically generate patient-friendly versions of reports alongside traditional technical ones. This dual-reporting approach would include clear explanations of each parameter, intuitive graphical representations of results within reference ranges, and contextual information about general health implications, all while carefully avoiding diagnostic conclusions.

The implementation of such an AI-guided interpretation system offers multiple advantages across the healthcare ecosystem. For patients, it provides unprecedented access to understandable information about their health status, fostering greater engagement in their healthcare journey. This improved comprehension can significantly reduce anxiety related to test results and decrease reliance on potentially misleading online sources. Healthcare professionals benefit from this system as well, as it allows them to focus their expertise on more complex aspects of patient care, such as developing comprehensive treatment plans and discussing nuanced clinical implications.

Perhaps most significantly, this system has the potential to transform the doctor–patient relationship. When patients arrive at consultations with a better understanding of their laboratory results, discussions can become more productive and focused on treatment decisions rather than basic result interpretation. This enhanced communication pathway creates a more collaborative healthcare environment, potentially leading to improved health

outcomes through better-informed decision-making and increased patient compliance with treatment plans.

### 4.1. Future Implementation and Integration

The long-term clinical utility of this AI interpretation system depends on seamless integration with existing laboratory information systems (LISs). Based on our preliminary technical assessment, integration could be achieved through three potential approaches:

1. **API-based integration**: Laboratory information systems could connect to the AI service via secure API calls, automatically sending structured report data for interpretation and receiving explanations that could be appended to standard reports or made available through patient portals. This approach would require minimal modification to existing LIS architecture and could be implemented with industry-standard HL7 FHIR protocols.

2. **Middleware solution**: For laboratories with legacy systems lacking API capabilities, a middleware layer could intercept report generation, process the data through the AI system, and reincorporate the interpretation before final delivery to patients or clinicians.

3. **Standalone patient-facing portal**: As an alternative requiring no direct LIS integration, laboratories could offer a secure portal where patients upload their reports for interpretation, though this approach would place more burden on users and potentially introduce transcription errors.

A critical consideration for any implementation approach is the mechanism for maintaining up-to-date reference ranges. We envision a dual-update system: (1) scheduled quarterly updates to incorporate systematic changes in laboratory methodologies or reference ranges, and (2) an alert-based system where participating laboratories can flag substantive changes requiring immediate AI system updates. The chatbot's prompts would include timestamp information to ensure transparency about when reference data were last updated.

Future research should focus on developing standardized formats for laboratory data exchange specifically designed for AI interpretation, potentially through extension of existing standards such as LOINC (Logical Observation Identifiers Names and Codes) to include AI-relevant contextual information. This aligns with Cabitza and Banfi's [9] vision of machine learning integration in laboratory medicine, where standardized data structures facilitate reliable algorithm performance.

### 4.2. Limitations

Despite the promising results, several important limitations of our study warrant consideration. First, while our data include reports from three different Italian laboratories, this regional focus may limit generalizability to laboratories with substantially different reporting standards, particularly those in countries with different units of measurement or reference range practices. International validation would be necessary before global implementation.

Second, although we included some pediatric cases (such as Case #5), our study is primarily focused on adult populations. Pediatric laboratory interpretation presents unique challenges due to rapidly changing developmental reference ranges and age-specific physiological considerations. Further research specifically targeting pediatric laboratory interpretation is needed.

Third, there exists a potential risk of patients misinterpreting AI-generated explanations as diagnostic advice, despite our careful design to avoid diagnostic language. In our patient feedback, 7% of participants indicated they might rely on the AI interpretation

without physician consultation for future results, highlighting the need for clear communication about the system's role as a supplement to, not replacement for, clinical consultation. Implementation would require careful user interface design with prominent disclaimers about the non-diagnostic nature of the interpretations.

Fourth, the perfect accuracy rate observed in our study may not be sustained in real-world implementation at scale, especially as laboratory methodologies evolve and new tests emerge. Ongoing monitoring and periodic revalidation would be essential for maintaining quality.

Finally, our evaluation focused on technical accuracy and patient comprehension but did not assess potential impacts on healthcare utilization or clinical outcomes. Longitudinal studies would be needed to determine whether improved laboratory result comprehension translates to measurable health benefits.

Regarding methodological aspects, we have implemented a three-stage training process: (1) initial familiarization with laboratory terminology and reference ranges, (2) supervised interpretation of increasingly complex cases, and (3) validation against expert interpretations. For values outside the training scope, we specifically engineered the system to acknowledge limitations with statements like "This parameter requires further evaluation by your healthcare provider as it falls outside common interpretation frameworks" rather than attempting speculative interpretations.

## 5. Conclusions

Our study represents a significant advancement in the application of AI technology to laboratory medicine, demonstrating that carefully designed AI systems can effectively bridge the gap between complex laboratory data and patient understanding. The Claude-based chatbot's ability to provide accurate, consistent, and patient-friendly interpretations across diverse laboratory parameters and reporting formats suggests a promising avenue for improving healthcare communication. While acknowledging the limitations of our current approach, we believe this research lays important groundwork for the broader integration of AI-assisted interpretation into clinical laboratory workflows.

The overwhelmingly positive patient feedback, coupled with the system's technical robustness, indicates that AI-enhanced laboratory reporting has the potential to transform patient engagement with healthcare data, improve doctor–patient communication, and ultimately contribute to more patient-centered care. Future research and implementation efforts should focus on addressing the identified limitations, particularly through international validation, special population studies, and longitudinal outcome assessments, while continuing to maintain the critical balance between accessible information and appropriate clinical governance.

This evolution in laboratory reporting represents not just a technological advancement, but a fundamental shift toward more transparent, accessible, and patient-empowering healthcare delivery. By providing patients with clear, accurate interpretations of their laboratory results, we can help them become more active participants in their healthcare journey, fostering better understanding, reduced anxiety, and more productive consultations with healthcare providers.

**Supplementary Materials:** The following supporting information can be downloaded at https://www.mdpi.com/article/10.3390/app15084232/s1. Figure S1: Supplementary figure showing an example of a prompt. Figures S2–S5: supplementary figures showing AI ChatBot interpretation of complete clinical chemistry analysis reports (Case #1, Case #4, Case #5, Case #6).

**Author Contributions:** Conceptualization, F.B.; methodology, F.D.; writing—original draft preparation, F.D. and F.B.; investigation, F.D., F.B., F.P. and M.P.; writing—review and editing, F.D. and F.B.; supervision, F.B. All authors have read and agreed to the published version of the manuscript.

**Funding:** This research received no external funding.

**Institutional Review Board Statement:** The study was conducted in accordance with the Declaration of Helsinki, and approved by the Ethics Committee of the University of Salento (protocol code AILAB-2024-01, approved on 15 November 2024).

**Informed Consent Statement:** Written informed consent has been obtained from the patients to publish this paper.

**Data Availability Statement:** Data is contained within the article.

**Conflicts of Interest:** The authors declare no conflicts of interest.

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
