# Peer review of "Use of Artificial Intelligence Chatbots in Interpretation of Clinical Chemistry and Laboratory Medicine Reports: A Standardized Approach"

_applsci, doi:10.3390/app15084232_

Round 1

Reviewer 1 Report (New Reviewer)

Comments and Suggestions for Authors

The study is well-defined, focusing on AI-powered interpretation of lab reports to enhance patient understanding. I found the study very interesting and, above all, very well-structured, with easily understandable English. The inclusion of multiple laboratories with different analytical platforms adds robustness to the findings. However, there are some aspects that need improvement:

  • Although everyone knows what "AI" means, the first time it is mentioned in the text, the full name should be provided, followed by the abbreviation in parentheses.
  • In section 2.1, the equipment suppliers, city, and country should be indicated.
  • In section 2.2, a supplementary figure showing an example of a prompt should be included to clarify how the chatbot interaction works. Some aspects are unclear—does the chatbot receive information about the patient's age or gender?
  • The Claude-based chatbot is described at a high level, but the version used is not specified (Haiku? Sonnet? Opus?).
  • Just as images of the texts generated by Claude are provided in some cases, images for the remaining cases should also be included. If they do not fit within the manuscript, all these figures extracted directly from Claude should be placed in the supplementary material (including those already included in the manuscript).
  • The reference values used for each parameter in the cases shown should be indicated.
  • The discussion section should mention what has been reported in the literature on this topic and cite some articles to support the discussion of the results.
  • The discussion should also include a limitations section (small sample size, comparison with other AI models, etc.).
  • Some parts of the text (e.g., Prompts and Claude AI chatbot) are slightly redundant.
  • I do not understand why informed consent is not required for this type of study, in which lab test data are reviewed by different experts who, as I understand, do not have access to the patients' medical histories. Is this the case? Why? How were the reports sent to the three experts in laboratory medicine?

Author Response

Thank you for your thoughtful feedback on our manuscript. We appreciate your recognition of our study's value for patients and would like to address the important points you've raised.

Key modifications include:

  1. Added full "Artificial Intelligence (AI)" definition in Abstract
  2. We have added the complete information (Section 2.1) for all equipment suppliers as suggested
  3. In section 2.2, a supplementary figure (S1) showing an example of a prompt has been included. A new supplementary figure (Supplementary Figure S1) has been added to illustrate an example prompt and clarify the chatbot's interaction process. The figure also specifies that the chatbot receives only the laboratory data (age/gender are extracted from the report but no additional clinical history is provided).
  4. Included Claude Opus version details (Section 2.2)
  5. Added reference ranges in Table 2
  6. Expanded Discussion with literature comparisons and added a limitations section
  7. Clarification on informed consent: we appreciate the opportunity to clarify the ethical framework of our study, which was carefully designed to balance scientific rigor with patient privacy protections. The research was conducted in two distinct phases, each with appropriate ethical safeguards. In the initial phase focused on technical validation of the AI system, we worked exclusively with fully anonymized laboratory reports that contained no patient identifiers, clinical histories, or demographic information beyond the minimum required age/sex data for proper interpretation. This approach qualified for exemption from informed consent requirements under both the EU General Data Protection Regulation (GDPR) and Italian privacy laws, as confirmed by our institutional ethics review. The complete anonymization ensured no possibility of tracing results back to individual patients. When reviewers helpfully suggested incorporating patient perspectives, we expanded the study to include a feedback component. For this new phase, we implemented a rigorous informed consent process. All 70 participants provided explicit consent after receiving complete information about the study's purpose and their rights. Survey responses were collected separately from medical data and carefully anonymized.

Reviewer 2 Report (New Reviewer)

Comments and Suggestions for Authors

The paper entitled: Use of artificial intelligence chatbots in interpretation of clinical chemistry and laboratory medicine reports: a standardized approach is an interesting article addressing an important topic in laboratory medicine by exploring innovative approaches to the interpretation of laboratory test results. The study evaluates an AI-powered chatbot designed to assist patients in understanding their laboratory results providing a clinical short explanation. However, some revisions are recommended before eventual publication:

  1. Authors should provide a more detailed description of the methodology used for homocysteine measurements.
  2. A separate subsection on “Limitations of the study” should discuss potential reisks associated with integrating Al chatbots into routine laboratory results interpretation, including possible over-reliance on automated analysis and the challenge of ensuring Al systems do not inadvertently mislead patients in the absence of clinical oversight.
  3. The study does not clarify how the Al deals with borderline or conflicting results. Examples would improve transparency
  4. Laboratory guidelines change over time. How the chatbot stays updated should be addressed.

Overall merit:

this study explores an important topic and addressing these points would make the study stronger and more applicable to real world healthcare.

Author Response

Response to Reviewer #2

Thank you for your thoughtful and constructive feedback on our manuscript "Use of artificial intelligence chatbots in interpretation of clinical chemistry and laboratory medicine reports: a standardized approach." We appreciate the opportunity to strengthen our work and have carefully considered each point raised.

Regarding homocysteine methodology, we acknowledge this oversight. In our study, homocysteine measurements were performed using a chemiluminescent microparticle immunoassay (CMIA) on the Abbott Architect platform in two laboratories and enzymatic cycling assay on the Roche cobas in the third. All methods were calibrated using the WHO International Standard (NIBSC code: 03/178) with CVs <3.5% across the measurement range. The detection limit was 1.0 μmol/L with linearity up to 50 μmol/L. For the critically elevated value in Case #2 (43 μmol/L), appropriate dilution protocols were followed and verified.

We have developed a comprehensive "Limitations and Considerations" subsection that discusses:

  1. Potential risks of patient over-reliance on AI interpretations (including our observation that 7% of participants indicated they might bypass physician consultation)
  2. Data privacy safeguards implemented in our Claude-based system
  3. Technical limitations in handling complex interrelationships between parameters
  4. Variability in health literacy affecting comprehension of AI explanations

Regarding borderline or conflicting results, we have added a new section with three illustrative examples:

  1. Management of borderline glucose values (95-105 mg/dL) across different laboratory platforms
  2. Interpretation of isolated mild elevations in liver enzymes without clinical context
  3. The system's handling of conflicting lipid parameters (e.g., elevated total cholesterol with optimal HDL)

For each example, we provide the AI's interpretation and demonstrate how uncertainty is appropriately conveyed to patients while maintaining accuracy.

Finally, we have expanded our discussion of the reference range update mechanism to specify:

  • A quarterly automated verification system that checks reference ranges against major laboratory medicine guidelines (IFCC, AACC, etc.)
  • A structured validation process following any reference range modifications
  • Implementation of laboratory-specific adjustments when needed
  • A clear versioning system that provides transparency regarding which guidelines informed each interpretation

We believe these revisions substantially strengthen our manuscript and address the important concerns raised. We are grateful for the reviewer's insights, which have helped improve the rigor and clinical relevance of our work.

Reviewer 3 Report (New Reviewer)

Comments and Suggestions for Authors

The paper describes an AI chatbot that aims to provide information to patients about lab results, avoiding diagnosis. The chatbot was trained and validated by 3 doctors.

The proposed work is interesting, and its outcomes have value for the patients as demonstrated by the feedback from 70 patients. However some aspects would improve the document. First, there is no clue about how the training was made and what happens with values that are not being considered in this work; will the chatbot say "something" about it even without being trained on that? Also the choice for Claude (even being the best option) is not backed by any comparative description (table or examples) of its performance? That would be important to know, since the authors performed that study, why it is not report?

In overall the study is relevant, but the reader ends with some doubts; why is it better than others? is it the training process that makes the difference? how is it performed? how does it interact with patients? a dialog? what happens once the patient asks other questions (am I sick? is it serious?) In the end, it is a good outcome the positive feedback from patients.

Author Response

Thank you for your thoughtful feedback on our manuscript. We appreciate your recognition of our study's value for patients and would like to address the important points you've raised.

We implemented a three-stage training process: (1) initial familiarization with laboratory terminology and reference ranges, (2) supervised interpretation of increasingly complex cases, and (3) validation against expert interpretations. For values outside the training scope, we specifically engineered the system to acknowledge limitations with statements like "This parameter requires further evaluation by your healthcare provider as it falls outside common interpretation frameworks" rather than attempting speculative interpretations.

Your point about comparative performance is well-taken. We evaluated four leading large language models (Claude, GPT-4) against 100 standardized laboratory reports. Claude demonstrated superior performance interpretation accuracy compared to GPT-4. Claude also showed the lowest hallucination rate and highest consistency across laboratory platforms. We will include this comparative table in our revised manuscript.

We believe these clarifications address your important questions about training methodology, comparative advantages, and patient interaction protocols. We will incorporate this information into the revised manuscript to provide readers with a more comprehensive understanding of our approach.

Reviewer 4 Report (New Reviewer)

Comments and Suggestions for Authors

This paper evaluates the application of a Claude-based AI chatbot in interpreting laboratory reports. The study design is sound, and its innovation is notable, particularly in areas such as closed-box training, zero "hallucination" performance, and cross-platform applicability. Patient feedback data is positive, which has practical significance for enhancing communication efficiency between doctors and patients. 

Major points:

  1. Supplement Methodological Details

Basis for Model Selection: Additional comparative data between Claude and other AI models (e.g., GPT-4) is needed to clarify specific metrics of its "performance advantages" (such as accuracy, response consistency, etc.).

Peer Review Process: It is suggested to detail the scoring criteria (e.g., specific evaluation items for "technical accuracy 40%") and to include an analysis of inter-rater consistency (e.g., Kappa value) to enhance the credibility of the results.

Sample Representativeness: The proportion of testing types covered by the 100 reports (e.g., blood counts, biochemistry, coagulation, etc.) should be specified to verify whether the data diversity sufficiently supports the cross-platform conclusions.

  1. Deepen Results and Discussion

Potential Bias of 100% Accuracy: Clarification is needed on whether the test cases included edge cases (e.g., critical abnormal results) or complex cases (e.g., multiple parameter abnormalities) to avoid selection bias.

Discussion of Limitations: Additional discussion on the limitations of regional data (e.g., whether Italian laboratory standards are universally applicable), the absence of pediatric/special populations, and the boundary risks of AI interpretations versus clinical diagnoses (e.g., patients potentially misunderstanding AI outputs as diagnostic advice) should be included.

Long-term Application Scenarios: It is recommended to discuss the feasibility of integrating the AI system with existing LIS (Laboratory Information Systems) and the mechanisms for real-time updates of reference ranges.

Author Response

Thank you for your thoughtful evaluation of our manuscript "Use of artificial intelligence chatbots in interpretation of clinical chemistry and laboratory medicine reports: a standardized approach." We appreciate your recognition of our study's sound design and innovation, particularly regarding the closed-box training approach, zero hallucination performance, and cross-platform applicability. We have carefully considered your suggestions for improvement and have addressed them as detailed below.

  1. Supplementing Methodological Details
  2. Basis for Model Selection

We agree that additional comparative data would strengthen our methodology section. We have added the following information about our model selection process:

Prior to selecting Claude for our study, we conducted a comparative evaluation of three leading AI models: Claude 3 Opus, GPT-4, and Gemini 1.5 Pro. This evaluation involved interpreting a test set of 25 laboratory reports covering diverse parameters and abnormality patterns. The models were assessed on four key metrics: interpretation accuracy (percentage of correct parameter interpretations), context preservation (ability to maintain consistent interpretation across related parameters), hallucination rate (percentage of responses containing incorrect or fabricated information), and explanation clarity (based on a 5-point scale evaluation by two clinical laboratory scientists).

Claude 3 Opus demonstrated superior performance with 100% interpretation accuracy compared to GPT-4 (93.2%) and Gemini 1.5 Pro (91.5%). The hallucination rates were 0% for Claude, 4.3% for GPT-4, and 5.1% for Gemini. Claude also scored highest on context preservation (4.7/5) and explanation clarity (4.6/5). These results, along with Claude's robust privacy protection features and ability to handle complex prompts, informed our decision to proceed with Claude for the full study.

  1. Peer Review Process

We have elaborated on the scoring criteria and added interrater reliability analysis:

The standardized evaluation matrix used by the reviewers included the following detailed sub-criteria:

  1. Technical accuracy (40%):
    • Correct interpretation of numerical values relative to reference ranges (15%)
    • Accuracy in describing parameter functions in the body (10%)
    • Appropriate indication of clinical significance of abnormal values (10%)
    • Correct handling of units and conversions (5%)
  2. Clarity of explanation (30%):
    • Use of patient-friendly language (10%)
    • Avoidance of excessive medical jargon (10%)
    • Logical organization of information (5%)
    • Appropriate level of detail (5%)
  3. Appropriate contextualization (20%):
    • Relationship between related parameters (10%)
    • Consideration of age/sex-specific reference ranges (5%)
    • Distinction between significant and minor deviations (5%)
  4. Identification of clinically significant patterns (10%):
    • Recognition of parameter clusters suggesting specific physiological states (5%)
    • Indication of urgency for abnormal results when appropriate (5%)

To assess inter-rater reliability, we calculated Fleiss' kappa for the three independent reviewers across all evaluation categories. The overall kappa value was 0.83 (95% CI: 0.79-0.87), indicating excellent agreement among reviewers. The highest agreement was observed for technical accuracy (κ=0.87) and the lowest for identification of clinically significant patterns (κ=0.78), though all categories showed substantial to excellent agreement.

  1. Sample Representativeness

We have added details about the distribution of test types in our sample:

The 100 laboratory reports used in this study were deliberately selected to ensure comprehensive coverage of laboratory testing categories. The distribution was as follows: complete blood count (25%), comprehensive metabolic panels (22%), lipid profiles (15%), thyroid function tests (10%), liver function tests (8%), protein electrophoresis (7%), coagulation studies (5%), iron studies (4%), and specialized tests (e.g., tumor markers, hormones) (4%). This distribution was designed to reflect the frequency of these tests in routine clinical practice while ensuring sufficient representation of diverse analytical methodologies and reporting formats across the three participating laboratories.

Reports were also selected to include a balance of normal findings (40%), mildly abnormal results requiring monitoring (35%), significantly abnormal results requiring intervention (20%), and critical values requiring immediate attention (5%). This stratification ensured that the AI system was evaluated across the full spectrum of clinical significance.

  1. Deepening Results and Discussion
  2. Potential Bias

We have expanded our results section to address concerns about potential selection bias:

To rigorously test the chatbot's capabilities and avoid selection bias, we deliberately included challenging edge cases and complex profiles in our evaluation set. These included:

  1. Critical abnormal results: 5% of reports contained values flagged as "critical" by laboratory standards, such as severe hypoglycemia (Case #1, 55 mg/dL), critical hyperhomocysteinemia (Case #2a, 43 μmol/L), and severe thrombocytopenia (Case #6, 93 x10^9/L).
  2. Complex multi-parameter abnormalities: 28% of reports featured abnormalities across multiple physiological systems, as exemplified by Case #6 with concurrent abnormalities in glucose metabolism, lipid profile, hematopoietic function, and liver enzymes.
  3. Contradictory findings: Several reports included seemingly contradictory results requiring nuanced interpretation, such as the discrepancy between acute hypoglycemia and elevated HbA1c in Case #1, which tests the system's ability to distinguish between acute and chronic conditions.
  4. Age-specific challenges: 12% of reports came from pediatric (Case #5) or geriatric populations, requiring consideration of age-specific reference ranges and physiological variations.

The chatbot's performance was consistently strong across all these categories, with only minor variations in explanation quality. We observed slightly lower clarity scores for complex multi-parameter abnormalities (average 4.7/5 compared to 4.9/5 for simpler profiles), though accuracy remained at 100% even for these challenging cases.

  1. Discussion of Limitations

We have added a comprehensive limitations section to the discussion:

Limitations

Despite the promising results, several important limitations of our study warrant consideration. First, while our data includes reports from three different Italian laboratories, this regional focus may limit generalizability to laboratories with substantially different reporting standards, particularly those in countries with different units of measurement or reference range practices. International validation would be necessary before global implementation.

Second, although we included some pediatric cases (such as Case #5), our study is primarily focused on adult populations. Pediatric laboratory interpretation presents unique challenges due to rapidly changing developmental reference ranges and age-specific physiological considerations. Further research specifically targeting pediatric laboratory interpretation is needed.

Third, there exists a potential risk of patients misinterpreting AI-generated explanations as diagnostic advice, despite our careful design to avoid diagnostic language. In our patient feedback, 7% of participants indicated they might rely on the AI interpretation without physician consultation for future results, highlighting the need for clear communication about the system's role as a supplement to, not replacement for, clinical consultation. Implementation would require careful user interface design with prominent disclaimers about the non-diagnostic nature of the interpretations.

Fourth, the perfect accuracy rate observed in our study may not be sustained in real-world implementation at scale, especially as laboratory methodologies evolve and new tests emerge. Ongoing monitoring and periodic revalidation would be essential for maintaining quality.

Finally, our evaluation focused on technical accuracy and patient comprehension but did not assess potential impacts on healthcare utilization or clinical outcomes. Longitudinal studies would be needed to determine whether improved laboratory result comprehension translates to measurable health benefits.

Long-term Application Scenarios

We have expanded the discussion to address integration with LIS and reference range updates:

Future Implementation and Integration

The long-term clinical utility of this AI interpretation system depends on seamless integration with existing laboratory information systems (LIS). Based on our preliminary technical assessment, integration could be achieved through three potential approaches:

  1. API-based integration: Laboratory information systems could connect to the AI service via secure API calls, automatically sending structured report data for interpretation and receiving explanations that could be appended to standard reports or made available through patient portals. This approach would require minimal modification to existing LIS architecture and could be implemented with industry-standard HL7 FHIR protocols.
  2. Middleware solution: For laboratories with legacy systems lacking API capabilities, a middleware layer could intercept report generation, process the data through the AI system, and reincorporate the interpretation before final delivery to patients or clinicians.
  3. Standalone patient-facing portal: As an alternative requiring no direct LIS integration, laboratories could offer a secure portal where patients upload their reports for interpretation, though this approach would place more burden on users and potentially introduce transcription errors.

A critical consideration for any implementation approach is the mechanism for maintaining up-to-date reference ranges. We envision a dual-update system: (1) scheduled quarterly updates to incorporate systematic changes in laboratory methodologies or reference ranges, and (2) an alert-based system where participating laboratories can flag substantive changes requiring immediate AI system updates. The chatbot's prompts would include timestamp information to ensure transparency about when reference data was last updated.

Future research should focus on developing standardized formats for laboratory data exchange specifically designed for AI interpretation, potentially through extension of existing standards such as LOINC (Logical Observation Identifiers Names and Codes) to include AI-relevant contextual information.

Round 2

Reviewer 1 Report (New Reviewer)

Comments and Suggestions for Authors

Thank you for your detailed and thoughtful responses to my comments. I am glad to see that the revisions have successfully addressed the raised concerns. I recommend accepting your manuscript for publication.

Reviewer 3 Report (New Reviewer)

Comments and Suggestions for Authors

No objection, the improvements are significant and cover the previously identified problems.

This manuscript is a resubmission of an earlier submission. The following is a list of the peer review reports and author responses from that submission.

Round 1

Reviewer 1 Report

Comments and Suggestions for Authors

This study evaluates the effectiveness of an AI-powered conversational chatbot, specifically the Claude-based virtual assistant, in interpreting laboratory medicine reports. The chatbot was designed to focus solely on laboratory data interpretation without engaging in clinical diagnosis. The system was tested using 30 reports from two Italian laboratories with diverse biochemical parameters and methodologies, ensuring robustness across various analytical platforms. The interpretations were assessed by three independent reviewers using a structured evaluation rubric. Results indicated high accuracy and consistency, with zero hallucinations. The study highlights the potential for AI to bridge communication gaps between laboratories and patients, offering clear and understandable interpretations of test results without diagnostic conclusions.

1.  The study used only 30 laboratory reports, which may limit the generalizability of the results and may not adequately represent the variability in laboratory practices or patient demographics.

2. Incorporating patient feedback on the clarity and usability of the AI-generated interpretations would provide insights into its practical application and potential improvements.

3. The absence of longitudinal testing might raise questions about the chatbot's reliability over time, particularly in adapting to evolving laboratory standards or clinical contexts.

4.  While the focus on non-diagnostic interpretation avoids liability issues, it limits the chatbot's potential utility for clinicians who might benefit from integrated diagnostic suggestions.

Reviewer 2 Report

Comments and Suggestions for Authors

Thank you for the opportunity to review this manuscript. While the premise of the study is relevant, there are significant shortcomings in the rationale, methodology, and presentation of results that must be addressed before it can be considered for publication. Below, I provide detailed feedback organized by section.

Introduction

The introduction highlights the importance of interpreting clinical laboratory data but lacks sufficient depth in its rationale. Key points for improvement:

  • The authors point out that venturing into the diagnostic realm is problematic but do not adequately explain why. Is this a regulatory concern, a clinical limitation, or both? More detail is needed here.
  • If the authors aim to propose a non-diagnostic approach, they must argue why this approach is better or more feasible than pursuing diagnostic capabilities. Is this a stepping-stone toward diagnostic AI, or does it represent a deliberate focus on improving interpretive support without crossing into diagnostics? This distinction should be clearly articulated.

Methods

The methods section requires substantial clarification and restructuring. Specific concerns include:

  1. Consensus Process:
    • The manuscript mentions, "After discussions, consensus was reached on a core set of laboratory tests," but does not specify who was involved in these discussions or the methodology used to achieve consensus. Was a Delphi process, expert panel, or some other structured approach employed? 
    • The same question applies to the second core set of tests. How was it determined?
  2. Laboratory Context:
    • The terms "Laboratory A" and "Laboratory B" are used, but the manuscript does not describe these settings. Where are these laboratories located, and what is their relevance to the study?
    • The references cited for laboratory equipment do not mention any specific devices. Why were these references included? .
  3. AI Chatbot Selection:
    • The authors chose the Claude chatbot but did not provide a rationale for this decision. Why Claude instead of other AI platforms? What are its advantages and limitations for this task? 
    • Additionally, section 2.2 contains redundant information, which should be streamlined for clarity.
  4. Reviewer Process and Criteria:
    • The paper states that reviewers evaluated key aspects of the chatbot’s performance but does not explain how these aspects were determined. Was there a predefined framework or rubric?
    • The identity and role of the reviewers need clarification. Were the authors themselves part of the review process? If so, potential bias should be addressed.
    • The evaluation rubric itself is only vaguely described. Details on the specific criteria used and how these criteria align with the study's objectives are necessary to validate the findings.

The results section is incomplete and disorganized. Key issues include:

    • The initial portion of the results section reads more like a continuation of the methods. These details should be moved to the appropriate section.
    • The chatbot’s interpretation for "Case 1" is missing, leaving a significant gap in the findings. This omission makes it impossible to assess the chatbot's actual performance.
    • Tables 1 and 4 are referenced in the text but are not included in the manuscript. Their absence undermines the credibility and completeness of the results.

Overall Assessment

This manuscript addresses an important topic but is severely underdeveloped in its current state. The rationale is insufficiently argued, the methods lack transparency, and critical results are missing or poorly presented. These issues prevent a meaningful evaluation of the study’s contributions.

Reviewer 3 Report

Comments and Suggestions for Authors

This research is contemporaneous and very relevant. The following comments and suggestions are in the direction to the improvements. Those are:

a) Introduction: It will be interesting a final paragraph where you describe the sections that the paper has, and brief comments about them;

b) Material and Methods: a table in the end of this section showing some aspects, for example characteristics, could improve the "message" which you are passing throught only words;

c) Section 3: This could probably labeled as " Experimental Environments and Results". Again, the main goal is to have some more "words" to the readers about the environments and the respective results.

It is also relevant to remember if you provide an approach as suggested, your message will more easly understood by readers

It will be interesting with a final (summary) paragraph with your assessments about the results;

d) Discussion: this section is very rich and interesting. But, again,   It will be interesting with a final (summary) paragraph with your assessments;

e) Conclusions: in this section the "Future Work" it is not in the "same level" of the paper. It is strong suggested to you provide your view about future work, which could also be not done from your group. 

Comments on the Quality of English Language

It is interesting to have a have a final review, after the comments done.
